# Crossing the Threshold: Idiomatic Machine Translation through Retrieval Augmentation and Loss Weighting

**Emmy Liu**    **Aditi Chaudhary** *    **Graham Neubig**
Carnegie Mellon University
emmy@cmu.edu, aditichaud@google.com, neubig@cs.cmu.edu

## Abstract

Idioms are common in everyday language, but often pose a challenge to translators because their meanings do not follow from the meanings of their parts. Despite significant advances, machine translation systems still struggle to translate idiomatic expressions. We provide a simple characterization of idiomatic translation and related issues. This allows us to conduct a synthetic experiment revealing a tipping point at which transformer-based machine translation models correctly default to idiomatic translations. To expand multilingual resources, we compile a dataset of $\sim$ 4k natural sentences containing idiomatic expressions in French, Finnish, and Japanese. To improve translation of natural idioms, we introduce two straightforward yet effective techniques: the strategic upweighting of training loss on potentially idiomatic sentences, and using retrieval-augmented models. This not only improves the accuracy of a strong pretrained MT model on idiomatic sentences by up to 13% in absolute accuracy, but also holds potential benefits for non-idiomatic sentences.[1]

## 1 Introduction

An idiom is a conventionalized expression in which the intended meaning differs from its literal translation. The translation of idioms has remained a problem for state-of-the-art research and commercial translation systems, as idioms tend to be translated literally (Dankers et al., 2022b; Shao et al., 2017; Anastasiou, 2010). Failure to translate these expressions correctly may lead to incomprehensible translations, particularly in literary text (Toral and Way, 2018). To illustrate the difficulty of understanding mistranslated idioms, we present mistranslations from commercial systems in Table 1.[3]

Although idiom translation has been recognized as a problem even before the advent of neural machine translation (Bar-Hillel, 1952; Wehrli, 1998), most work has focused on identifying and evaluating the problem cross-linguistically (Baziotis et al., 2022; Dankers et al., 2022b), or on interpreting the behaviour of transformer-based models in translating or memorizing idioms (Haviv et al., 2022; Dankers et al., 2022b). Others pose idiom identification and paraphrasing as a separate task from machine translation (Pershina et al., 2015). Comparatively fewer recent works have attempted to remedy this problem. Early work made use of idiom dictionaries and direct substitution, or example-based machine translation (Salton et al., 2014; Nagao, 1984). However, we would ideally want to make use of the contextual translation abilities of neural models. Data augmentation and the creation of new datasets have helped address this problem (Agrawal et al., 2018), but it may also be possible to use existing data resources more effectively, especially for higher-resource languages.

We first frame the general problem of non-compositional translation, which encompasses the translation of idioms and other multi-word expressions that cannot be translated word-for-word (§2). We then perform synthetic experiments in a very simple case, finding that transformer-based machine translation models generally translate word-for-word until a proportional threshold of sentences contain non-compositional expressions, at which point the translations flip to being correct (§4.1). We evaluate translations by commercial models in three natural languages, and find a drop in performance on idiomatic sentences and stronger performance on more common idioms (§4.2). We hypothesize that this may reflect similar trends as exist in processing other long-tail phenomena, and similar tactics to those used to deal with rare phenomena may work (Kandpal et al., 2022).

With this intuition, we improve the idiomatic

---

* Currently works at Google Research.

[1]Code and data available at https://github.com/nightingal3/idiom-translation/

[3]Translations from commercial systems were collected at the end of 2022.

| Source | Target | Translation | Language | System |
|---|---|---|---|---|
| Vous devez avoir la dalle. | You gotta be starving. | You must have the slab. | `fr` | DeepL |
| Il lui faut toujours chercher la petite bête. | He has to dot all the i's, cross all the t's. | He always has to look for the little beast. | `fr` | DeepL |
| J'ai la pêche à mort. | Good as hell. | I have the peach to death. | `fr` | Google |
| 弱肉強食 | The Weak are Meat, the Strong do Eat. | The Weak are the Strong. | `ja` | DeepL |
| その手は食わないわ | Oh, no, I'm not falling for that. | I'm not gonna eat that hand. | `ja` | DeepL |
| 知らぬが仏って事もある | Well, sometimes what we don't know doesn't hurt us, right? | I don't know, but sometimes Buddha | `ja` | Google |
| Eukko elää kuin pellossa. | Whoa. Homegirl clearly never met a trashcan. | The hen lives like a field. | `fi` | DeepL |
| Sinun olisi pitänyt ottaa minut mukaan tähän isoon päätökseesi - tavasta, jolla isämme heittää lusikan nurkkaan. | You should have included me in this huge decision you made about how our father's gonna leave this Earth. | You should have included me in this big decision of yours - the way our father throws the spoon in the corner. | `fi` | DeepL |
| Roger voi tykätä kyttyrää. | Roger may take a dim view of this... | Roger may like a hunchback. | `fi` | Google |

Table 1: Examples of mistranslated sentences produced by commercial translation systems. Idioms and their corresponding translations are highlighted in red.[2]

translations generated by a strong pretrained machine translation model, $\Delta$LM (Ma et al., 2021), without harming the translation quality of literal expressions. To contribute resources toward documenting idioms and improving their translation cross-linguistically, we create a dataset of sentences containing idiomatic expressions in three languages (French (`fr`), Finnish (`fi`) and Japanese (`ja`) (§3). We propose two simple but effective ways to improve translation of idioms, namely upweighting training loss on potentially idiomatic sentences and retrieval augmentation (§5). We find that this can improve the idiomatic translation abilities of the model significantly, by an average of 10.4% in absolute accuracy (§7.1). Moreover, this does not harm translation of sentences where the literal sense of the idiom is used, and it improves translation of out-of-distribution sentences in French and Finnish as well. We perform human evaluation and error analysis, and find that the rate of severe semantic errors is reduced by an average of 7.52% absolute accuracy (§7.2). The ultimate aim for machine translation is to ensure accessibility for all texts. This requires addressing idiomatic phrases, culturally-informed language, and complex semantics. We demonstrate the potential for enhancing idiom translation using existing resources.

## 2 Non-Compositional Translation

### 2.1 Background on Idioms

Idioms are commonly understood to be fixed expressions that contradict the principle of compositionality in language, which is to say that their meaning cannot be predicted from the meanings of their parts (Radford, 2004; Portner, 2005). Idioms occur relatively frequently in all languages, and are often challenging for non-native speakers (Cooper, 1999). For instance, a literal translation of one Portuguese idiom is "*it is from little that you twist the cucumber*". This is difficult to understand. However, an equivalent English expression is "As the twig is bent, so is the tree inclined", which refers to actions during childhood influencing behaviours that people have as adults (Unbabel, 2019). This example illustrates the importance of translating idioms using equivalent idioms from the target culture, or a paraphrase if there is no equivalent.

Idiomatic expressions are heavily shaped by the culture of language speakers, including religious beliefs, history, geography, and cuisine. For instance, food-related idioms in English tend to refer to foods such as beef and potatoes, while in Chinese, these idioms tend to refer more to rice and tofu (Yang, 2010). Cross-cultural knowledge is important in choosing a translation that conveys the proper intent to readers in the target language (Liu, 2012). Overly-literal translations and lack of broader context are two reasons why machine translation is still not at parity with human translators, particularly when translating literary text (Matusov, 2019; Omar and Gomaa, 2020; Poibeau, 2022).

### 2.2 Formal definition

We use the idea of non-compositionality to frame idiomatic translation more precisely. Let $\mathcal{X} = \{x_1, ..., x_N\}$ be the set of tokens in the source language, and $\mathcal{Y} = \{y_1, ..., y_M\}$ be the set of tokens in the target language. Suppose that we have an oracle function TRANSLATE : $\mathcal{X}^* \to \mathcal{Y}^*$ that always produces a correct translation. We can imagine this to be a helpful speaker who is perfectly familiar with both languages and never misreads text. Then we can say that a multi-token string

requires non-compositional translation if it can be translated correctly by the oracle as a whole, but it cannot be translated correctly by individually translating parts of the sentence and joining them (according to the target language's word order). In other words, for a string of tokens $x_1, ..., x_n$, [4]

$$\bigoplus_{i=1}^{n}{}_Y \text{TRANSLATE}(x_i) \neq \text{TRANSLATE}(\bigoplus_{i=1}^{n}{}_X x_i) \quad (1)$$

We note that this definition is very general and also includes other phenomena such as multi-word expressions and named entities. However, we can now use this definition to create a relevant synthetic task, allowing us to observe translation compositionality under different settings (§4.1).

## 3 Idioms and Data Collection

We can use the formal definition from the previous section to generate synthetic data for experiments. However, we ultimately want to improve translation of real idioms. To do so, we collect a dataset of natural sentences to evaluate commercial systems and the model we seek to improve.

Although a large corpus of potentially idiomatic expressions exists in English (Haagsma et al., 2020), there are no readily accessible equivalents in other languages. Therefore, we collected idioms in French, Finnish, and Japanese from language-learning sites, listed in Appendix B. These languages were chosen for phylogenetic diversity, and due to availability of commercial translation systems. In total, there were 148 French idioms collected, 92 in Finnish, and 1336 in Japanese.

To collect sentences containing these idioms, we matched on lemmatized forms from the 2018 version of OpenSubtitles (Lison et al., 2018), where lemmatization was performed with Stanza (Qi et al., 2020). In total, there were 85632 French sentences containing potentially idiomatic expressions, 51811 Finnish sentences, and 23018 Japanese sentences. To filter out unaligned sentences, we scored each source and reference sentence using COMET-QE (Rei et al., 2020) and removed the bottom 10% of each language's sentences by COMET-QE scores.

Some idioms have a plausible literal meaning (such as "kick the bucket" to mean kicking a physical bucket). To make sure that all examples in

the idiomatic test set were actually idiomatic, we sorted sentences into an idiomatic test set where the idiomatic meaning of a phrase was used (e.g. "to die") and a literal test set, where the literal meaning of the phrase was used (e.g. kicking a physical bucket). The first 100 examples containing each idiom's lemmatized form were collected, and up to the first 3 (for Japanese) or 5 (for Finnish and French) literal and figurative examples in this set were collected to create the test set. This was to avoid dominance of very common idioms in the test set. This created two test sets related to the idiom list for each language, the *idiomatic* and *literal* test sets.

To validate these judgments, we hired native annotators in French and Finnish. They were presented with examples from the final literal and idiomatic test sets in a shuffled order, and asked to label them with idiomatic, literal, or N/A labels if they didn't think it was an instance of either. Agreement (Krippendorff's $\alpha$ (Krippendorff, 1970; Castro, 2017)) in both cases was moderately high (French $\alpha = 0.5754$, Finnish $\alpha = 0.6454$). Details can be found in Appendix D.

Finally, we collect two random test sets, one which is in-domain and another which is out-of-domain. For the in-domain test set, we simply select sentences from the development set of Open-Subtitles (see subsection 6.2 for details on our split of OpenSubtitles). For the out-of-domain test set, we use the Ted Talks corpus (Reimers and Gurevych, 2020). This is to ensure that translation quality of other, unrelated sentences is not impacted by any modifications meant to improve translation of idioms. Topics discussed and vocabulary used in Ted Talks may be slightly different from what is discussed in movies or TV shows, so training the model on OpenSubtitles and testing on Ted Talks allows us to evaluate model generalization. For both test sets, to control for translation length as a source of difficulty, sentences were length-matched on the target side with corresponding sentences in the idiomatic set. This created the *random* set, which is the same size as the idiomatic test set. All three test sets are summarized in Table 2.

## 4 Evaluating Non-Compositional Translation

### 4.1 Artificial Language Translation

We first use the definition of non-compositional translation in (§2) to create a synthetic task. This

---

[4] $\bigoplus_X$ denotes string concatenation given the word order of language $X$, i.e. if the word order is SVO, the tokens belonging to the subject should be placed in front of the tokens belonging to the verb, and so on.

| Language | Idiom matches | Idiomatic | Literal | Random (in) | Random (out) | Total |
|----------|---------------|-----------|---------|-------------|--------------|-------|
| fr | 85632 | 777 | 79 | 777 | 777 | 2410 |
| fi | 51811 | 449 | 81 | 449 | 449 | 1428 |
| ja | 23018 | 3253 | 389 | 3253 | 3253 | 10148 |

Table 2: Size of test sets for each language. The idiomatic and literal sentences contain strings matching known idioms (after lemmatization), and the in-domain random set contains unrelated sentences from Open-Subtitles, but the out-of-domain random set contains unrelated sentences from the Ted Talks corpus.

allows us to gain an understanding of how much data is required to memorize non-compositional patterns. Although this experiment is not realistic to natural language (notably, there is no token-level ambiguity in this experiment), we note that using synthetic experiments allows us to easily extend the data generation setup and examine model behaviour along many different conditions, such as informativity.

The source language in these experiments was composed of tokens 0 through 9, $X = \{0, 1, 2, ..., 9\}$. The target language was produced by adding 10 to each token, $Y = \{10, ..., 19\}$. The translation rule was to add 10 to the value of each token in the source language, e.g. $0 \rightarrow 10$, $1 \rightarrow 11$. We add a single non-compositional rule that doesn't follow this trend, $0\ 1 \rightarrow 12$ (rather than $0\ 1 \rightarrow 10\ 11$). We limited the maximum sequence length to 6 tokens.

We generated synthetic training corpora of several sizes containing different numbers of occurrences of the non-compositional rule $0\ 1 \rightarrow 12$. The number of training sentences ranged from 100k to 10M, while the number of noncompositional occurrences ranged from 10 to 1M. We examined two informativity conditions, corresponding to the case where the context provides no information (tokens are randomized around the non-compositional expression), and the context being perfectly informative. The perfect informativity condition was achieved by adding the canary token "11" to the source vocabulary, and only inserting this token prior to the non-compositional pattern "0 1".

We experimented with three different transformer sizes (Vaswani et al., 2017), each of which had a hidden dimension and embedding size of 512, as well as 16 attention heads. Only the number of encoder and decoder layers varied, such that the small transformer had 3 encoder and decoder layers, the medium transformer 8, and the large transformer 16. We fix the number of epochs for the small, medium and large models to respectively

be 10, 20, and 30 in the non-informative case and 15, 15 and 25 in the informative case.[5] Further training details can be found in Appendix A.

Although this may seem like a simple task, we found it surprisingly difficult for models to learn this non-compositional pattern. Results in each setting, averaged across 5 random seeds, are presented in Figure 1. Especially for the small model, there is a sharp gradation from translating none of the non-compositional expressions correctly to translating them all correctly, which occurs when roughly 10% of training data contains a non-compositional pattern. A similar trend exists for larger models, but the threshold is less distinct. This corroborates the tendency for transformers to translate non-compositional phrases literally (Dankers et al., 2022b). Comparatively less data is required when the context is informative, but the trends remain similar to the non-informative case. As model size and corpus size increase, the rate of correct translations for non-compositional examples actually drops, contrary to expectation.

It is unlikely that any individual idioms occur in 10% of sentences in natural language. Due to the highly regular translation rules in this synthetic language, there may be a stronger bias toward translating compositionally in this experiment. However, we gain the intuition that idioms can be translated effectively if they appear frequently, and that clear context clues reduce data required.

## 4.2 Evaluation of Commercial Systems

Although synthetic experiments provide intuition on the difficulty of translating idioms, one might ask whether similar results hold in natural language. To answer this, we examine the performance of commercial systems on the test sets in (§3). Namely, we examine Google Translate and DeepL on Finnish, French, and Japanese idiomatic, literal, and random sentences. Results are in Table 3. We observe drops in translation quality on idiomatic sentences in all languages, with lower automatic metrics overall.

---

[5]The number of training epochs was determined by the number of epochs it took for the validation loss to plateau in the 100k size corpus with 1k non-compositional examples, rounded up to multiples of 5. This was done to mimic the typical training process for MT models, which are trained until loss or accuracy plateaus on a general dev set. Since idiomatic expressions tend to be uncommon compared to literal ones, there may not be many in the dev or train sets, and so the model's performance on idiomatic expressions may not be tracked.

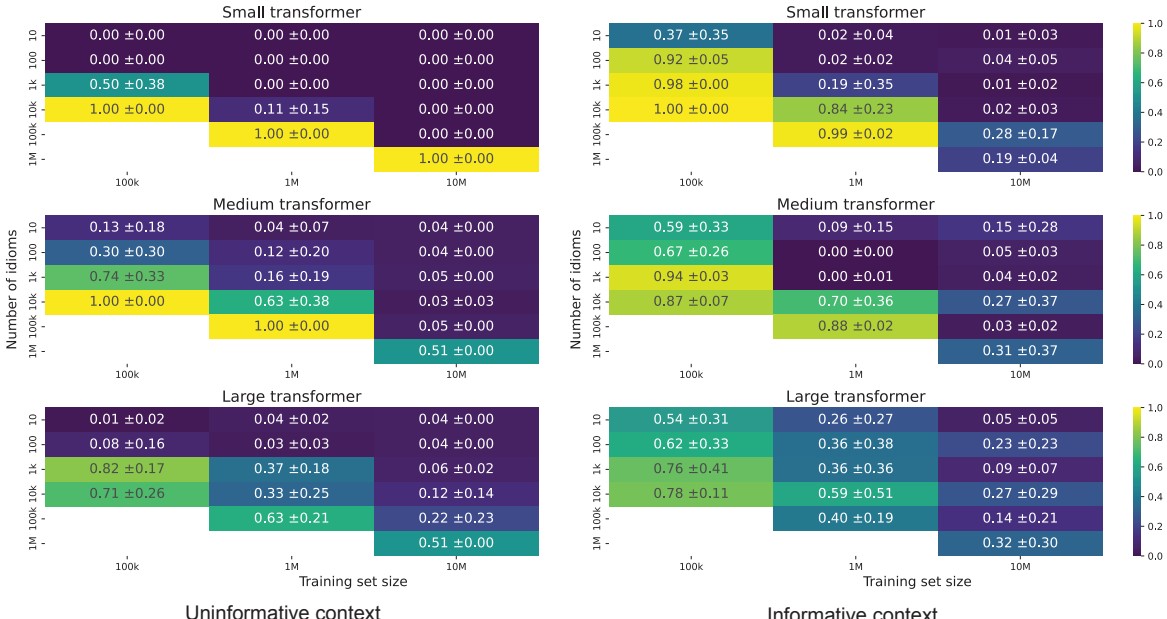

Figure 1: Accuracy of a transformer in translating a non-compositional phrase after training on datasets of different sizes, with different numbers of non-compositional patterns (only non-compositional translation accuracy is depicted). Results are averaged across 5 seeds, and standard deviation is shown.

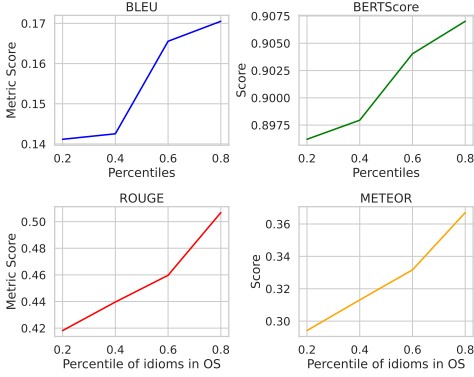

Figure 2: Automatic metrics – Quality of DeepL French translations on idiomatic test set bucketed by idiom frequency. The bottom 20% of least common idioms are excluded, as they may occur fewer than 3 times and not be in our test set.

Although it's impossible for us to determine what data these commercial systems were trained on, we examine the frequency of each idiom within OpenSubtitles as a proxy for its overall frequency in the training data, and bucket idioms into quintiles based on their occurrence frequency in source text. As idioms become more frequent, the quality of translations increases. An example of DeepL on the French idiom set is shown in Figure 2. Trends for other languages and systems are in Appendix H. This indicates that like in the synthetic experiments, there may be strong frequency effects on translation quality of idioms.

| Language | System | BLEU | METEOR | BERTScore |
|---|---|---|---|---|
| fi$_{\text{idiomatic}}$ | DeepL | 0.1001 | 0.2497 | 0.8866 |
| | Google | 0.0923 | 0.2250 | 0.8726 |
| | ΔLM-base | 0.1608 | 0.3592 | 0.9126 |
| fi$_{\text{literal}}$ | DeepL | 0.1488 | 0.3908 | 0.9146 |
| | Google | 0.1398 | 0.3577 | 0.9017 |
| | ΔLM-base | 0.2093 | 0.5050 | 0.9350 |
| fi$_{\text{random-out}}$ | DeepL | 0.2052 | 0.4082 | 0.9103 |
| | Google | 0.2288 | 0.4357 | 0.9062 |
| | ΔLM-base | 0.2365 | 0.4971 | 0.9145 |
| fr$_{\text{idiomatic}}$ | DeepL | 0.1575 | 0.3278 | 0.9006 |
| | Google | 0.1261 | 0.2794 | 0.8808 |
| | ΔLM-base | 0.2001 | 0.4393 | 0.9211 |
| fr$_{\text{literal}}$ | DeepL | 0.2219 | 0.4022 | 0.9122 |
| | Google | 0.2034 | 0.3830 | 0.9012 |
| | ΔLM-base | 0.2778 | 0.5504 | 0.9377 |
| fr$_{\text{random-out}}$ | DeepL | 0.2854 | 0.4650 | 0.9125 |
| | Google | 0.3103 | 0.4922 | 0.9149 |
| | ΔLM-base | 0.2778 | 0.5504 | 0.9377 |
| ja$_{\text{idiomatic}}$ | DeepL | 0.1172 | 0.2735 | 0.8932 |
| | Google | 0.0672 | 0.1839 | 0.8644 |
| | ΔLM-base | 0.09048 | 0.2998 | 0.9234 |
| ja$_{\text{literal}}$ | DeepL | 0.1517 | 0.3440 | 0.9059 |
| | Google | 0.0937 | 0.2565 | 0.8829 |
| | ΔLM-base | 0.1416 | 0.4222 | 0.9222 |
| ja$_{\text{random-out}}$ | DeepL | 0.1074 | 0.2934 | 0.8878 |
| | Google | 0.1079 | 0.2834 | 0.8829 |
| | ΔLM-base | 0.0948 | 0.3436 | 0.8946 |

Table 3: Performance of commercial systems on idiomatic, literal, and random test sets. There is a clear degradation in performance on idiomatic sentences.

## 5 Methods to Improve Non-Compositional Translation

We explore two methods to improve translation, loss weighting and kNN-MT. These two methods are relatively simple to use, where loss weighting

only requires a list of potentially idiomatic phrases in the source language, and kNN-MT only requires enough space on disk to save the datastores.

More formally, we consider the basic case of autoregressive machine translation, with a set of parallel sentences in the source ($X = \{x^{(i)}\}_{i=1}^{N}$) and target ($Y = \{y^{(i)}\}_{i=1}^{N}$) language: $\mathcal{D} = \{(x^{(i)}, y^{(i)}), ..., (x^{(N)}, y^{(N)})\}$. The model $p_\theta$ with parameters $\theta$ is trained by minimizing the loss:

$$\mathcal{L}(\theta, \mathcal{D}) = \sum_{i=1}^{N} \ell(y^{(i)}, p_\theta(x^{(i)})) \qquad (2)$$

**Upweighting** here refers to sentence-level up-weighting, where there is a set of sentences $A$ that we'd like to upweight with a weight coefficient $\alpha$. In this case, $A$ would be potentially idiomatic sentences. We keep all other parameters for training the same as in the base model.

$$\mathcal{L}(\theta, \mathcal{D}) = \sum_{i=1}^{N} \alpha^{\mathbb{1}(x^{(i)} \in A)} \ell(y^{(i)}, p_\theta(x^{(i)})) \qquad (3)$$

**kNN-MT** augments a translation model with a retrieval component (Khandelwal et al., 2021). Given each sentence $(x, y)$, we construct a data-store with keys based on hidden representations constructed from the translation model, and values being the next word in the target sentence.

During generation, a probability distribution over next words can be computed based on the retrieved next words and the distance of their keys to the current context. A parameter $\lambda$ controls interpolation between the distribution over next words predicted by the base model, and the distribution predicted by the retrieved $k$ neighbours.[6]

$$p(y_i^{(j)}|x^{(j)}, y_{1:i-1}^{(\hat{j})}) = \lambda p_{\text{kNN}}(y^{(j)}|x^{(j)}, y_{1:i-1}^{(\hat{j})}) + (1-\lambda)p_\theta(y_i^{(j)}|x^{(j)}, \hat{y}_{1:i-1}^{(j)}) \qquad (4)$$

We also combine loss weighting with kNN-MT, where a model is trained with sentence upweighting and interpolated with a datastore based on representations from the upweight-trained model.

Intuitively, these methods make sense to use for idiom translation – we have previously seen that one problem with non-compositional phrases may

---

[6]We run a hyperparameter search using the validation set to find the best kNN-MT settings for each language. Further details are in Appendix C.

simply be their rarity. Upweighting training examples that contain idioms may help with under-representation. Furthermore, retrieving similar examples may find occurrences of the same idiom which were translated correctly.

## 6 Experimental Settings

### 6.1 Experimental Settings

We run experiments on $\Delta$LM-base, a transformer encoder-decoder model with 360M parameters, a larger version of which ranked first in the WMT21 multilingual translation task (Ma et al., 2021; on Machine Translation , WMT21). We train one $\Delta$LM model for each language pair. Each model was trained for 2 million steps, and the checkpoint with the best loss on the validation set was kept. Further details are in Appendix C. To decode, we used beam search with a beam size of 5.

### 6.2 Data

Models were trained on OpenSubtitles for each language pair. Data from test sets were removed, and 10% of the remaining data was used as a validation set. There were 33.8M sentences in the `fr-en` train set, 22.0M in `fi-en`, and 1.6M in `ja-en`.

### 6.3 Evaluation

We use multiple automatic metrics to evaluate translation quality. However, due to the importance of accurate semantic evaluation, the authors (native English speakers and fluent in French and Japanese) conduct a human evaluation inspired by MQM (Lommel et al., 2014). Only errors that would fall under the "terminology" and "accuracy" error types are considered, as we are focused on severe semantic errors. We give a score of 0 for severe errors and a score of 0.5 for major errors. A score of 1 is given otherwise. Exact evaluation standards are in Appendix E.

## 7 Results

### 7.1 Automatic and Human Evaluation

In most cases, as reported in Figure 3, using a combination of sentence upweighting and kNN-MT led to the greatest increase in automatic metrics on all three test sets, of up to 3.08 BLEU points on the idiomatic test set (`fr`), 2.69 BLEU points on the literal test set (`fi`), and 5.75 points on the random test set (`fr`). In all cases except `ja-rand`, using one or more of these methods improved over

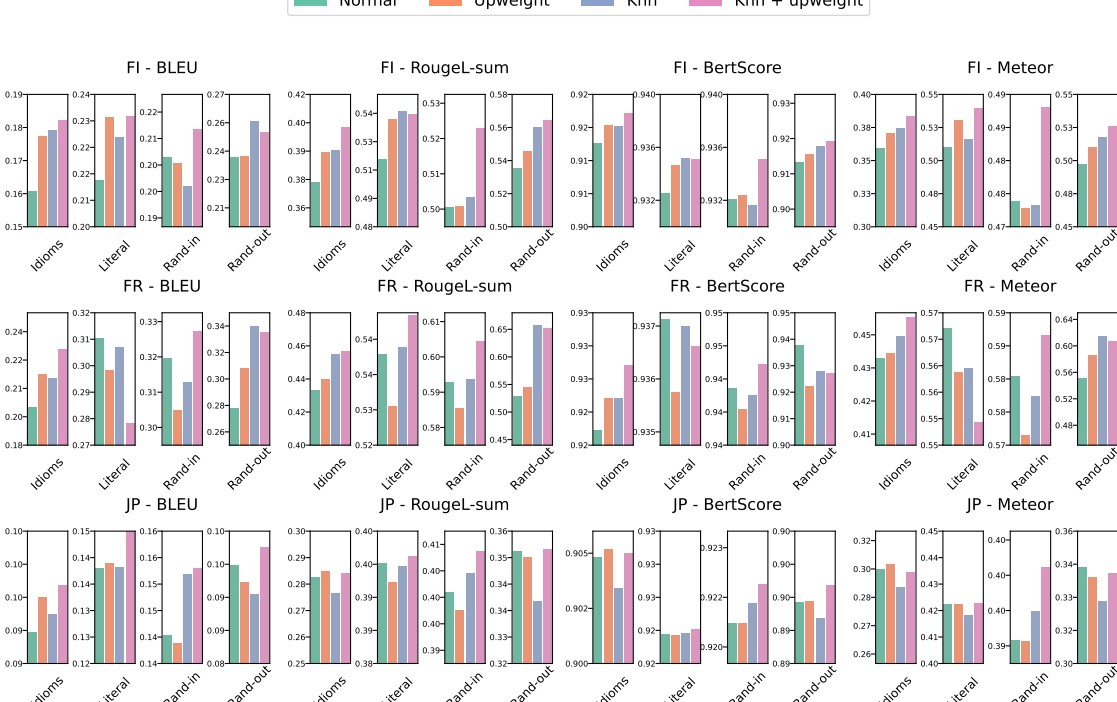

Figure 3: Results of automatic metrics. In most cases, combining loss weighting with KNN-MT improves automatic metrics the most on all three test sets, including the out-of-distribution (Random) test set.

the baseline. Exact numerical results are in Appendix J.

We evaluate the statistical significance of the results through a one-tailed permutation test (Graham et al., 2014). Further details are in Appendix F. Exact results are in Appendix G. For Finnish, significance is achieved for all three test sets, and for French, significance is achieved for the idiomatic and random test sets. For Japanese, values achieved are not significant, but are borderline.

As our focus is on mitigating semantic errors, we mostly focus on the results of human evaluation, which are summarized in Table 4. Here, we also find that using both sentence upweighting and kNN is the best condition in most cases, increasing accuracy by roughly 13% in French and Finnish, and 4.5% in Japanese for idiomatic sentences. Encouragingly, this does not overly harm translation of literal sentences, as accuracy on the literal set either increases slightly (by roughly 4% in French and Finnish), or decreases very slightly (by roughly 0.4% in Japanese). For the random set, the combination of sentence upweighting and kNN-MT by around 7% accuracy. However, in Japanese, performance on the random test set decreases by 4%. In all cases except `ja-rand`, one or more of these methods improves over the baseline.

We note that the Japanese model was trained on roughly 1/10th of the data of the French and Finnish models, so its translations are not as high-quality. This also leads to the construction of a much smaller datastore, which may lead to weaker performance on the random set.

| | base | knn | upweight | upweight + knn |
|---|---|---|---|---|
| `fr-idioms` | 0.6177 | 0.6659 | 0.7010 | **0.7463** |
| `fr-literal` | 0.7039 | 0.7303 | 0.7105 | **0.7434** |
| `fr-rand-out` | 0.7526 | **0.8398** | 0.7477 | 0.8232 |
| `fi-idioms` | 0.4803 | 0.5562 | 0.5604 | **0.6194** |
| `fi-literal` | 0.7692 | **0.8462** | 0.8205 | 0.8141 |
| `fi-rand-out` | 0.7647 | 0.8235 | 0.7771 | **0.828** |
| `ja-idioms` | 0.4152 | 0.4286 | **0.4643** | 0.4598 |
| `ja-literal` | 0.6475 | 0.6516 | **0.6557** | 0.6434 |
| `ja-rand-out` | **0.6207** | 0.5560 | 0.5776 | 0.5862 |

Table 4: Human-judged accuracy on sentence-level semantics.

## 7.2 Error analysis

We repeat the frequency analysis performed on commercial systems (§4.2) for ΔLM, and find that adding upweighting and kNN-MT generally improves translations at all frequency levels. These increases are not concentrated in low-frequency idioms, so more common idioms continue to be translated better.[7] A representative example (for

---

[7]This trend is different in retrieval of long-tail facts in question answering, in which retrieval flattens out the difference

French) is in Figure 4. A complete set of plots are in Appendix I.

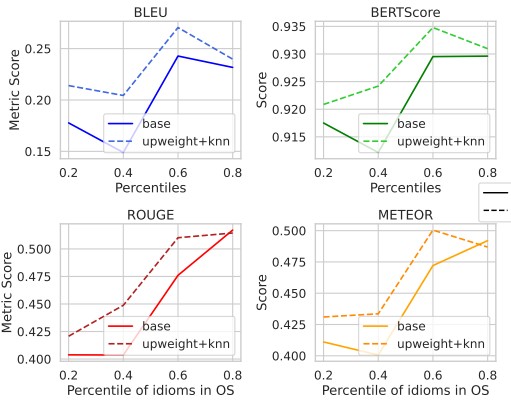

Figure 4: Automatic metrics for `fr-idiom` sentences, plotted by frequency, for base and upweight+knn.

We examine the rate of severe and major errors made in the base model and the upweight+knn model in Table 5. In French and Finnish, the rate of critical errors decreased greatly, particularly in the idiomatic and random test sets. This is true to a lesser extent in Japanese. Major errors also decreased to a lesser extent. The only test set where errors increase is again the `ja-rand` test set. We note that it's possible for the rate of major errors to be higher in the upweight+knn model because some severe errors transitioned to major errors.

| | System | Severe (↓) | Major (↓) |
|---|---|---|---|
| fi-idioms | base | 0.4258 | 0.1648 |
| | upweight+knn | **0.3242** | **0.0962** |
| fi-literal | base | 0.1728 | **0.1234** |
| | upweight+knn | **0.1234** | 0.1358 |
| fi-random | base | 0.1317 | **0.2009** |
| | upweight+knn | **0.0603** | 0.2188 |
| fr-idioms | base | 0.3042 | 0.1528 |
| | upweight+knn | **0.198** | **0.1092** |
| fr-literal | base | 0.2326 | 0.1047 |
| | upweight+knn | **0.2209** | **0.04651** |
| fr-random | base | 0.1624 | 0.1688 |
| | upweight+knn | **0.09407** | **0.1649** |
| ja-idioms | base | 0.4643 | 0.2411 |
| | upweight+knn | **0.4464** | **0.1875** |
| ja-literal | base | 0.2867 | **0.1311** |
| | upweight+knn | **0.2787** | 0.1557 |
| ja-random | base | **0.2931** | **0.1724** |
| | upweight+knn | 0.3190 | 0.1897 |

Table 5: Rate of major and severe errors in translations.

One question is why the error rate on out-of-distribution sentences drops for French and Finnish. In `fi-rand`, the severe error rate more than halves (0.1317 → 0.603), and in `fr-rand`, it nearly halves (0.1624 → 0.09407). However, it is

between rare and common facts (Kandpal et al., 2022).

unclear why this should be the case. We examined sentences where the original translation was incorrect but the upweight+knn translation was correct, and found that they tended to contain named entities. For instance, for the sentence "*La chirurgie à coeur ouvert au Nigeria, c'est un gros problème.* (Open heart surgery in Nigeria - big trouble.)", the base model incorrectly produced the translation "Open-heart surgery in Forbes, that's a big problem.", while the upweight+knn model translated correctly. In some cases, words with multiple possible translations (e.g. *spectre*: ghost, spectrum) became correctly translated. "*Mais regardez le nombre de lignes noires dans ce spectre.* (But look at the number of black lines in that spectrum.)" was originally translated incorrectly as "But look at the number of black lines in that ghost".

## 8 Related Work

Recent work has raised the issue of idiom handling in MT (Baziotis et al., 2022; Dankers et al., 2022b,a). There is historical recognition of the problem, including of multi-word expressions (Sag et al., 2002; Calzolari et al., 2002; Zaninello and Birch, 2020). This has historically motivated example-based machine translation (Nagao, 1984). Similar motivations underlie the use of kNN-MT. However, neural models may already be capable of translating idiomatic phrases if they appear often enough in training data.

Other works focus on data augmentation and creating new data resources (Ho et al., 2014; Fadaee et al., 2018; Agrawal et al., 2018; Haagsma et al., 2020). A related task is detection of conventionalized metaphors (Levin et al., 2014). Automatic identification of idiomatic phrases, as well as data augmentation are promising avenues to improve performance in lower-resource languages.

Instance weighting has been explored previously in the MT literature, but has been mostly explored in the context of domain adaptation, rather than being used to improve translations of rare or non-compositional phrases in the same domain (Foster et al., 2010; Wang et al., 2017).

Idiomatic phrases are a prototypical case of phrases that need to be memorized (Haviv et al., 2022). Many also occur infrequently in training data, which may make it difficult for transformer-based models to translate them (Kandpal et al., 2022). This can be mitigated, as we have shown in this paper. However, more work is needed to effec-

tively learn idioms and other infrequent linguistic elements with few repetitions.

## 9 Conclusion

We highlight the challenge idiomatic expressions pose to machine translation systems and provide simple solutions to improve performance. Through synthetic experiments, we identify a threshold at which transformer-based models correctly default to idiomatic translations. We develop a dataset of sentences containing idiomatic expressions in French, Finnish, and Japanese, and introduce two techniques - upweighting training loss on potentially idiomatic sentences and augmenting models with kNN-MT - which enhance the idiomatic translation accuracy of a strong model, while offering potential benefits for non-idiomatic sentences.

Future research could extend these techniques to additional languages, and explore their effectiveness in dealing with other long-tail phenomena. We hope that this work contributes toward increasing the intelligibility of translations containing idioms or set phrases. Ultimately, for machine translation to be useful for everyone without causing misunderstandings, "last mile" problems involving cultural knowledge, long-tail phenomena, and complex semantic evaluation should be taken into account.

## Acknowledgements

Thank you to Perez Ogayo for helping with DeltaLM setup, all annotators who validated idiomatic/literal judgments, as well as to NeuLab members for providing feedback on parts of the draft. This project was funded by the P2020 program MAIA (LISBOA-01-0247-FEDER-045909).

## 10 Limitations

Our research provides a first step toward capturing non-compositional expressions in machine translation. However, we do not conclusively solve the problem, as ideally a machine translation system should be able to learn any idiom or non-compositional phrase from a few examples.

First, our experiments were conducted on a select group of languages (Finnish, French, and Japanese), which do not fully capture the variety and complexity of languages worldwide. Given the diversity of language structures and idiomatic expressions, the generality of our findings to languages with drastically different grammatical structures or idiom usage patterns remains uncertain.

Next is our use of synthetic data. While synthetic data allowed us to control for certain variables, our setting is purposefully simplified, potentially limiting the ecological validity of our findings. Although our synthetic language was designed to mimic non-compositional translation issues, it may not encapsulate the full extent of such complexities in real-world languages. Namely, there is only one non-compositional pattern and the remaining translations are one-to-one mappings.

Our research also depends on the quality and representativeness of the training and evaluation corpora. For instance, certain idioms may be overrepresented or underrepresented, which could affect the translation performance.

Lastly, our improvement methods, namely upweighting and kNN-MT, have inherent limitations. Upweighting could lead to overfitting on idiomatic expressions and may not be as effective when idioms occur infrequently in the data. On the other hand, kNN-MT might not yield significant improvements if the idiom or its correct translation rarely appears in the training data, limiting its utility in such scenarios.

Future work could address these limitations by expanding the linguistic scope of the study, exploring more complex methods or architectures, or investigating to what extent similar techniques can be applied to related issues in semantic preservation during machine translation.

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

## A  Synthetic Dataset Training Details

Three encoder/decoder transformer sizes were trained, differing only in number of encoder/decoder layers. The small size had 3 encoder/decoder layers, the medium size had 8 encoder/decoder layers, and the large size had 16 encoder/decoder layers. For all models, the hidden dimension was 512, the embedding dimension was 512, and there were 16 attention heads.

In the experiments without informative context, the small transformer was trained for 10 epochs, the medium transformer for 20, and the large for 30. In experiments with context, this was changed to 15, 15, and 25 respectively. These values were

based on early experimentation with loss plateaus on the validation set.

Sentences in the synthetic dataset were composed of tokens as described in subsection 4.1. Sentences were constrained to be 1-6 tokens in length.

Experiments with the synthetic dataset were implemented in PyTorch.

## B Idiom Sources

Sources of idioms were pulled from language-learning websites below:

`fi`:

- https://en.wiktionary.org/wiki/Appendix:Finnish_idioms

`fr`:

- https://speechling.com/blog/20-most-common-french-idioms-to-get-you-talking-like-a-native/

- https://frenchtogether.com/french-idioms/

- https://vidalingua.com/blog/funny-french-idioms-explained-english

`ja`:

- https://www.theintrepidguide.com/japanese-expressions-and-idioms/

- https://en.wikipedia.org/wiki/Japanese_proverbs

- https://www.fluentu.com/blog/japanese/japanese-idioms-2/

- https://kotowaza.jitenon.jp/

## C OpenSubtitles Training Details

A separate `deltalm-base` was trained from the pretrained model for 2M steps on each language. The Adam optimizer was used (Kingma and Ba, 2017), with a learning rate of 1e-4, betas of (0.9, 0.98), and an inverse square root learning rate scheduler with 4000 warmup updates (with a warmup learning rate of 1e-7, and a minimum learning rate of 1e-9). The maximum number of tokens in a batch was set to 1024, with maximum source and target lengths of 512. Label smoothing of 0.1 was used, and the loss function used was cross-entropy.

For upweighting experiments, the upweight coefficient for each language was found through a hyperparameter search on the validation set over $\alpha \in \{3, 5, 10, 50\}$. The values for each language were $\alpha_{fi} = 3, \alpha_{fr} = 10, \alpha_{ja} = 3$.

For kNN-MT, three datastores were built for approximate kNN search using the training set from OpenSubtitles. These datastores were built with the `faiss` library (Johnson et al., 2019). The Finnish datastore contained 248M vectors with 507k centroids, while the French datastore contained 348M with 713k centroids, and the Japanese datastore 17.8M with 73k centroids. All vectors were stored in `fp16` with a code size of 32. The vectors used as keys in the datastore correspond to the *input* to the last feedforward layer. Additionally, a hyperparameter search for each language was carried out on the validation set, over values of $\lambda \in \{0.2, 0.4, 0.6, 0.8\}$, temperature $\in \{0.1, 1, 10\}$, and number of retrieved neighbours $\in \{5, 10, 15, 20\}$. Hyperparameters were selected based on BLEU score on the validation set.

The best hyperparameters were as follows: `fi`: ($\lambda = 0.4$, temp = 10, num neighbours = 20), `fr`: ($\lambda = 0.2$, temp = 10, num neighbours = 20), `ja`: ($\lambda = 0.4$, temp = 10, num neighbours = 20) During test time, the best hyperparameters for each language were used, with a probe size of 20 and beam size of 5.

Experiments with ΔLM and kNN-MT were implemented in fairseq (version from September 2021) (Ott et al., 2019).

## D Native Annotator Recruiting

Annotators were hired through Upwork, and consisted of professional translators in French and Finnish who were also fluent in English. Annotators were paid between $10 - 25$USD an hour, depending on their individual hourly rate. No personally identifying information was collected.

Annotators were shown the task description below, and also had access to a compiled list of literal and figurative translations for each idiom.

**Description:**

This job is to create a dataset that will allow us (researchers at Carnegie Mellon University, PI Graham Neubig) to study the ability of machine translation systems to translate idioms.

We would like you to look at translations and decide whether they are literal or idiomatic translations. These will be English sentences containing the translation of a sentence in either French, Finnish, or Japanese. All the sentences contain a phrase that is an idiom in the foreign language, but sometimes the translations may be meant literally. As an example, in English "kick the bucket" can mean to die when used in an idiomatic way, or to literally kick a bucket when used in a literal way. You should mark the example with "idiomatic" if you think it's an idiomatic example, or "literal" if you think it's a literal example.

If you think the translation is not a good translation, we will ask you to mark this entry with the word "None", but this should occur in relatively few cases.

We estimate that the job will take 2-3 hours overall.

Examples:

Original sentence: Jalkaväkeä oli pilvin pimein. English translation (reference): Arty and infantry just kept on coming. Contains idiom: pilvin pimein Idiom literal meaning: In dark clouds Idiom figurative meaning: A huge (often excessive) amount of something. Label: idiomatic

Original sentence: Harmi, ettei lehdistötilaisuudessasi ollut lammasta. English translation (reference): Too bad they didn't have lamb at your press conference, Francis. Contains idiom: olla lammas Idiom literal meaning: To be a lamb Idiom figurative meaning: a person who is like a lamb does nothing alone. the person does Label: literal

## E  Standards for Human Evaluation

Evaluation standards were based on MQM standards, in particular the major and critical severity levels, which are defined below (Lommel et al., 2014):

- **Major severity error**: severity level of an error that seriously affects the understandability, reliability, or usability of the content for its intended purpose or hinders the proper use of the product or service due to a significant loss or change in meaning or because the error appears in a highly visible or important part of the content.

- **Critical severity level**: severity level of an error that renders the entire content unfit for purpose or poses the risk for serious physical, financial, or reputational harm.

We slightly adapted this for idioms, where if it was possible to infer the meaning of a sentence through the existence of a similar English idiom, but it would not be something generally said by English speakers, we assigned a major severity. Due to ΔLM often making errors with numbers and named entities, we assigned these errors a major rather than critical severity in most cases, although these would generally be critical severity errors

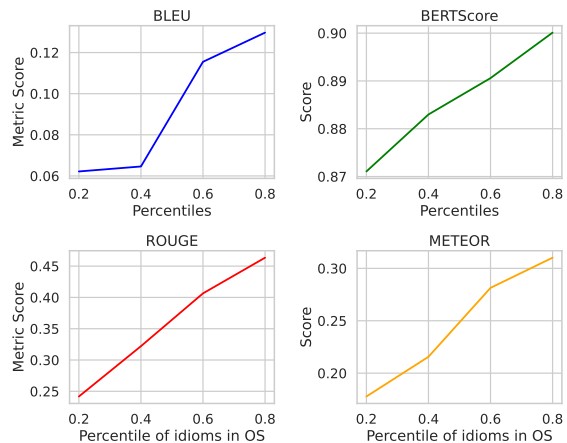

Figure 5: Effect of idiom frequency on translations by DeepL in Finnish

in business documents. If part of a sentence was missing, we based the error severity on whether or not the missing portion was crucial to the intent of the sentence. Examples of sentences labelled with error severity are provided in Table 6:

## F  Statistical Significance Testing

For each language and test set, we examine the null hypothesis that the BLEU scores of the two systems actually have the same distribution, and the difference occurred by chance.

For each source sentence, we shuffled the translations produced by the two systems with probability 0.5, and then recalculated BLEU scores. This was repeated 1000 times, and the number of times the shuffled difference was greater or equal to the observed difference was recorded to find the p-value.

## G  Statistical Significance Results

Statistical significance results are shown in Table 7.

## H  Effect of Frequency on Commercial Translations

The effect of idiom frequency on commercial models' translations is in Figure 5 through Figure 9.

## I  Effect of Frequency on ΔLM

The effect of frequency on ΔLM for Finnish and Japanese is shown in Figure 10 and Figure 11

## J  Automatic Metrics in Detail

Exact results for automatic metrics are shown in Table 8.

| Language | Src | Ref | Hyp | Severity |
|---|---|---|---|---|
| fr | - Eh oui, l'habit ne fait pas le moine. | Yes, clothes don't make the man. | Yes, clothes don't make you a monk. | Major |
| | Il t'a fait une queue de poisson sur l'autoroute. | He cut you off on the freeway. | He made you a fish tail on the highway. | Severe |
| | Il pleut des cordes. | It's raining cats and dogs. | It's raining. | Major |
| fi | Ei pane tikkuakaan ristiin. | He does nothing. | Doesn't even cross a match. | Severe |
| | Tämä meni yli hilseen. | This is just way over my head. | This is over the top. | Major |
| | Hieno aasinsilta itsemurhaan. | Nice speech, Mr. Hobson. Way to hit the suicide theme. | It's a nice morning for suicide. | Severe |
| ja | 意表を突くってことを 君は学ぶ必要がある | You need to learn the element of surprise. | You need to learn to play hard-to-get. | Severe |
| | もうその手は食わないわ | I'm not falling for this shit. | I won't have to do that again. | Major |
| | 一杯食わせやがったな！ | You swindled me! | You've given me enough to eat! | Severe |

Table 6: Examples of categorized errors made by $\Delta$LM.

| Language | Test set | p-val |
|---|---|---|
| fi | idioms | 0.0* |
| | literal | 0.003* |
| | rand | 0.018* |
| fr | idioms | 0.0* |
| | literal | 0.234 |
| | rand | 0.0* |
| ja | idioms | 0.058 |
| | literal | 0.083 |
| | rand | 0.088 |

Table 7: p-values obtained using approximate randomization on translations produced by the base model and the upweight+knn model.

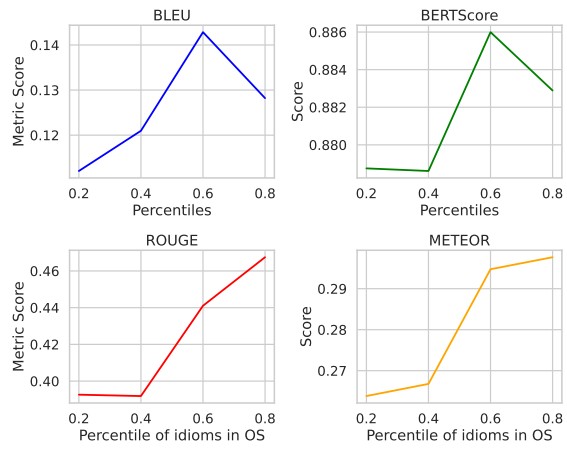

Figure 7: Effect of idiom frequency on translations by Google in French

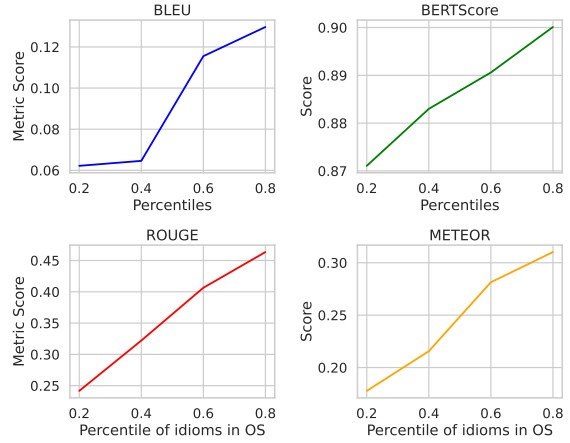

Figure 6: Effect of idiom frequency on translations by Google in Finnish

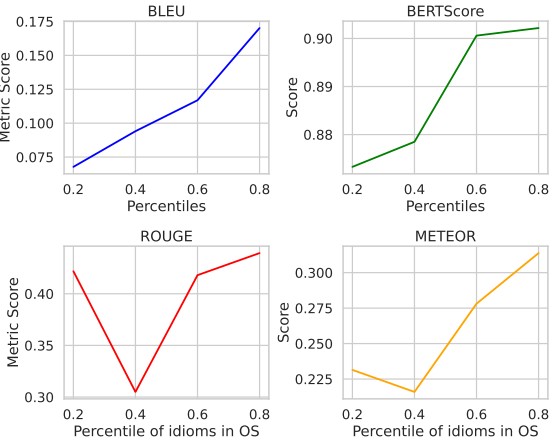

Figure 8: Effect of idiom frequency on translations by DeepL in Japanese

| Language | Test Set | Model | BLEU | RougeL-sum | BertScore | Meteor |
|---|---|---|---|---|---|---|
| fi | Idioms | Normal | 0.1608 | 0.3737 | 0.9126 | 0.3592 |
| | | Knn | 0.179 | 0.3904 | 0.9152 | 0.3745 |
| | | Upweight | 0.1773 | 0.3895 | 0.9154 | 0.3709 |
| | | Knn + upweight | **0.182** | **0.4027** | **0.9172** | **0.3833** |
| | Literal | Normal | 0.2093 | 0.5053 | 0.9350 | 0.5050 |
| | | Knn | 0.2257 | 0.535 | 0.937 | 0.5161 |
| | | Upweight | 0.2314 | 0.5281 | 0.9364 | 0.5258 |
| | | Knn + upweight | **0.2362** | **0.5335** | **0.9387** | **0.5217** |
| | Random-in | Normal | 0.2056 | 0.5044 | 0.9321 | 0.4739 |
| | | Knn | 0.1991 | 0.5067 | 0.9316 | 0.4732 |
| | | Upweight | 0.2044 | 0.5048 | 0.9324 | 0.4728 |
| | | Knn + upweight | **0.2120** | **0.5224** | **0.9351** | **0.488** |
| | Random-out | Normal | 0.2365 | 0.535 | 0.9145 | 0.4971 |
| | | Knn | **0.2557** | 0.5602 | 0.9183 | 0.5178 |
| | | Upweight | 0.2374 | 0.5458 | 0.9163 | 0.5102 |
| | | Knn + upweight | 0.2498 | **0.564** | **0.9193** | **0.5261** |
| fr | Idioms | Normal | 0.2001 | 0.4329 | 0.9211 | 0.4393 |
| | | Knn | 0.2154 | 0.4547 | 0.9235 | 0.4493 |
| | | Upweight | 0.2174 | 0.4398 | 0.9235 | 0.4419 |
| | | Knn + upweight | **0.2309** | **0.4568** | **0.926** | **0.4581** |
| | Literal | Normal | 0.2778 | 0.5261 | 0.9377 | 0.5504 |
| | | Knn | 0.2824 | 0.5318 | 0.9387 | 0.5544 |
| | | Upweight | **0.2923** | **0.5332** | 0.9384 | **0.5564** |
| | | Knn + upweight | 0.2883 | 0.5325 | **0.9399** | 0.5549 |
| | Random-in | Normal | 0.3197 | 0.5942 | 0.9443 | 0.5803 |
| | | Knn | 0.3142 | 0.5949 | 0.9438 | 0.5774 |
| | | Upweight | 0.3079 | 0.5884 | 0.9427 | 0.5715 |
| | | Knn + upweight | **0.3258** | **0.6036** | **0.9461** | **0.5865** |
| | Random-out | Normal | 0.2778 | 0.528 | 0.9377 | 0.5504 |
| | | Knn | **0.3396** | **0.6573** | **0.928** | **0.6146** |
| | | Upweight | 0.3078 | 0.5458 | 0.9223 | 0.5856 |
| | | Knn + upweight | 0.3353 | 0.652 | 0.9272 | 0.6075 |
| ja | Idioms | Normal | 0.09048 | 0.2826 | 0.9234 | 0.2998 |
| | | Knn | 0.09376 | 0.2767 | 0.9034 | 0.2905 |
| | | Upweight | 0.09505 | **0.285** | **0.9052** | **0.3022** |
| | | Knn + upweight | **0.09589** | 0.2841 | 0.905 | 0.2982 |
| | Literal | Normal | 0.1416 | 0.3951 | 0.9222 | 0.4222 |
| | | Knn | 0.1418 | 0.3947 | 0.9223 | 0.4183 |
| | | Upweight | 0.1427 | 0.3923 | 0.9221 | 0.4222 |
| | | Knn + upweight | **0.1509** | **0.3962** | **0.9226** | **0.4228** |
| | Random-in | Normal | 0.1443 | 0.4008 | 0.9212 | 0.3927 |
| | | Knn | 0.1535 | 0.4036 | 0.9218 | 0.3960 |
| | | Upweight | 0.1430 | 0.3981 | 0.9212 | 0.3925 |
| | | Knn + upweight | **0.1543** | **0.407** | **0.9224** | **0.4009** |
| | Random-out | Normal | 0.0948 | 0.3539 | 0.8946 | **0.3436** |
| | | Knn | 0.09051 | 0.3389 | 0.8934 | 0.3284 |
| | | Upweight | 0.09228 | 0.3522 | 0.8947 | 0.3392 |
| | | Knn + upweight | **0.09752** | **0.3545** | **0.8959** | 0.341 |

Table 8: Results of automatic metrics. In most cases, combining loss weighting with KNN-MT improves automatic metrics the most on all three test sets, including the out-of-distribution (Random) test set.

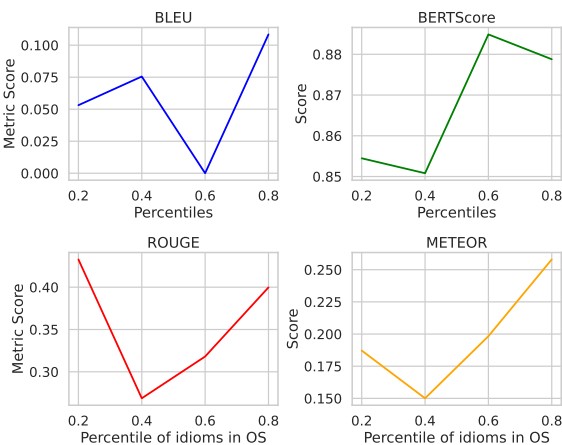

Figure 9: Effect of idiom frequency on translations by Google in Japanese

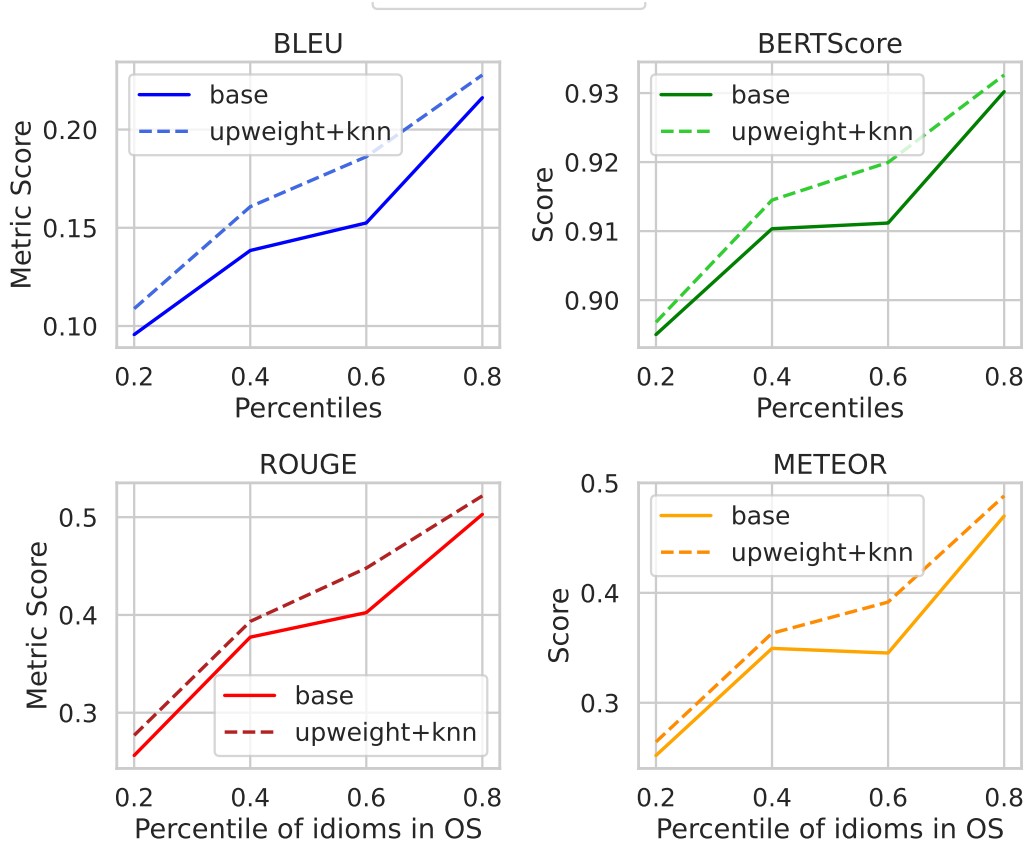

Figure 10: Translation metrics for Finnish idiomatic sentences, for base and upweight+knn models.

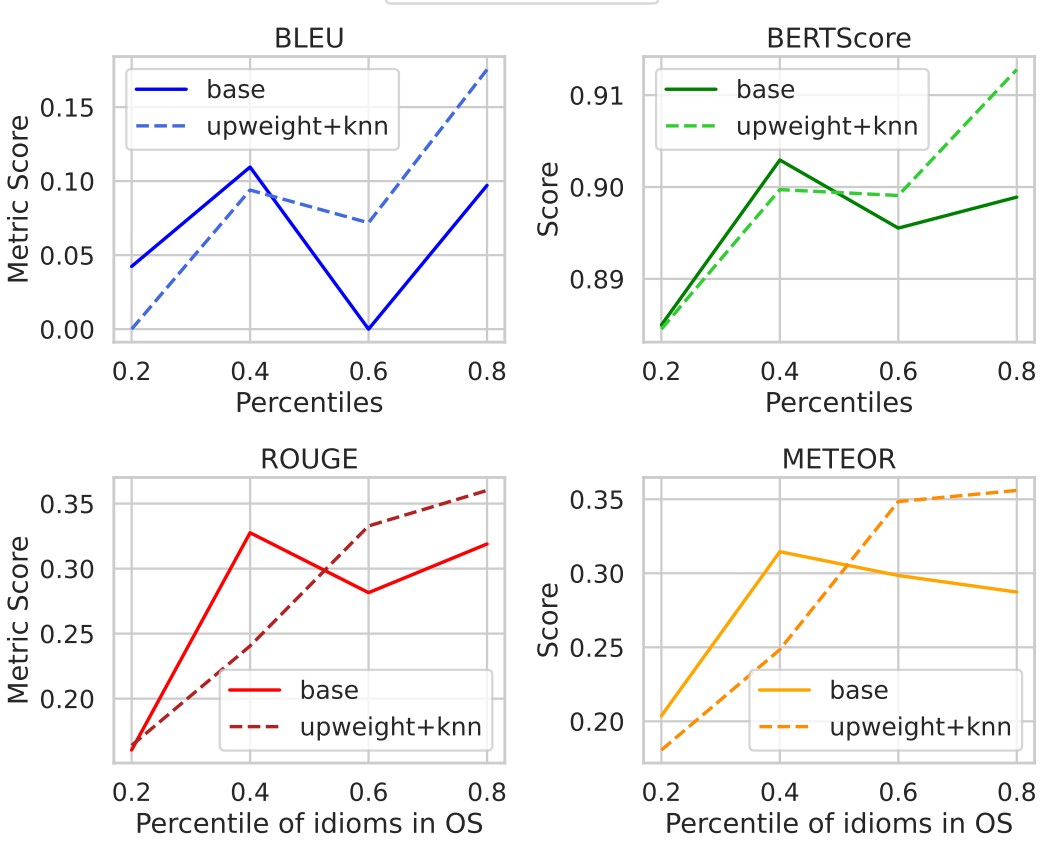

Figure 11: Translation metrics for Japanese idiomatic sentences, for base and upweight+knn models.