# OpenReview forum: "Crossing the Threshold: Idiomatic Machine Translation through Retrieval Augmentation and Loss Weighting"
_EMNLP/2023/Conference — EMNLP 2023 Main_

### Official Review · Reviewer_23HH · 2023-08-04

**Soundness:** 4

**Excitement:**

3: Ambivalent: It has merits (e.g., it reports state-of-the-art results, the idea is nice), but there are key weaknesses (e.g., it describes incremental work), and it can significantly benefit from another round of revision. However, I won't object to accepting it if my co-reviewers champion it.

**Paper Topic And Main Contributions:**

The paper on idiom translation is a well-written and well executed piece of research. The authors present a comprehensive study that delves into the impact of idiom phrase frequency in training data on model learning. The synthetic evaluation proposed by the authors sheds light on an essential aspect of idiom translation. By examining how the frequency of idiom phrases influences the model's ability to translate them accurately, the authors offer valuable insights into the intricacies of idiomatic language processing.
They incorporate two existing methods on their datasets to see if they can improve the translation quality of idioms. The experiments conducted in the study are sound and provide strong evidence to support the conclusions drawn by the authors.


**Questions For The Authors:**

- Clarification on the synthetic setup.

**Reasons To Accept:**

- Idiom translation is an open question in the field.
- Their proposed approach of using upweight in addition to knn-MT seem to be working quite well.
- The paper introduces a new dataset for idiom translation for 3 languages.

**Reasons To Reject:**

- The analysis on synthetic translation experiments don't seem valid to me. The authors assume a specific ordering in translated words as well as a paradigm where every word is translated into one other word. I'm suspecting what they analyze in such a case is not non-compositionality in language but a specific and simple case of pattern recognition.

- While the paper introduces new datasets, the novelty of the work is not acceptable for me. The methods used in this work have already been existing and applying them to a new problem, while useful, might not have enough contribution for a long paper.

**Reproducibility:**

3: Could reproduce the results with some difficulty. The settings of parameters are underspecified or subjectively determined; the training/evaluation data are not widely available.

**Reviewer Confidence:**

4: Quite sure. I tried to check the important points carefully. It's unlikely, though conceivable, that I missed something that should affect my ratings.

---

> ### Author Rebuttal · Authors · 2023-08-29
>
> Hello, thank you for your review, we’re glad that you appreciate the results of the paper. We wrote a general response as well, and pasted relevant sections here. See below for responses to individual questions and comments as well:
>
> ### General response
>
> We thank all reviewers for their detailed responses and suggestions. We plan to update our manuscript based on some suggestions raised, and we also respond to some common concerns below:
>
> 2. Applicability of the synthetic experiment
>
> Several reviewers also raised concerns about the applicability of the synthetic experiment to real data. We would like to reiterate that this is meant to be a simple experiment by design and is not meant to capture the complexity of idioms in natural language. The motivation behind this experiment was to provide a simple testing ground so that factors such as corpus size, idiom frequency, sentence length, and informativity of the context could be easily manipulated. Past analysis in this area has focused on real idioms already (Shao et al 2018, Dankers et al. 2022), but this is based on a few examples, and it is not as easy to manipulate natural language data. This experiment was meant to be unambiguous and simple in order to complement the other two experiments conducted on natural language data.
>
> We agree that further extensions can be made in order to make the setting more realistic, including manipulating informativity conditions further, creating multiple non-compositional rules and longer non-compositional ngrams, and manipulating the occurrence rate of tokens in non-compositional phrases outside the phrase. Below are responses to individual questions and comments:
>
> ### Comments
>
> > The analysis on synthetic translation experiments don't seem valid to me. The authors assume a specific ordering in translated words as well as a paradigm where every word is translated into one other word. I'm suspecting what they analyze in such a case is not non-compositionality in language but a specific and simple case of pattern recognition.
>
> Yes, we agree that non-compositionality in language is actually more of a spectrum, and the synthetic experiment in fact is meant to test a case of pattern recognition. Namely, the recognition of a non-compositional example when compositional translations are more common. We do state that this is meant to be a simple case, and is a complement to the later experiments with real idioms.
>
> > While the paper introduces new datasets, the novelty of the work is not acceptable for me. The methods used in this work have already been existing and applying them to a new problem, while useful, might not have enough contribution for a long paper.
>
> Although we agree that the methods that we use already existed and had been applied in other settings, we think that the contribution may still provide value to the community. Namely, idiom translation has been a problem for a long time, and providing some simple methods that boost performance is still of value to the community if it has not been done before.
>
> ### Answers to Questions
>
> > Clarification on the synthetic setup.
>
> See general response, item 2.

---

### Official Review · Reviewer_dV8h · 2023-08-04

**Soundness:** 2

**Excitement:**

2: Mediocre: This paper makes marginal contributions (vs non-contemporaneous work), so I would rather not see it in the conference.

**Paper Topic And Main Contributions:**

This paper addresses the challenging task of idiom translation, and performs multiple experiments related to that:
1. Firstly, an experiment is conducted that uses synthetically generated data of strings of numbers, to examine what proportion of non-compositional expressions is required for the system to be able to learn that non-compositional translation. An example 'sentence' would be "0 3 4 7 0 1" --> "10 13 14 17 12", where "0 1" are the non-compositional mapping. Transformer models of varying sizes are trained corpora of varying sizes. Approximately 10% of non-compositional data was needed for the models to learn the non-compositional mapping.
2. A new dataset is created by taking French, Finnish and Japanese idioms, and matching them with Opensubtitles sentences. The sentences are sorted into literal and figurative examples, using 3-5 examples per idiom, for literal and idiomatic occurrences.
3. Using the newly created dataset, two commercial translation systems (DeepL and Google Translate) and one LM (LM-base) are evaluated, which provides convincing evidence that these systems, too, have subpar performance on idiomatic inputs (when compared to literal occurrences of PIEs and random data from the TED talks corpus). Idioms that occurred more frequently in Opensubtitles also have better translations.
4. Finally, the paper proposes two modelling techniques that can improve idiom processing. 1) Upweighting sentences that contain PIEs, 2) knn-MT. For Finnish and French clear improvements are observed for the two methods, and the two methods combined. For Japanese, the results are a bit less consistent: combining the two typically performs similar to normal training, but only applying one of the two slightly decreases performance. Across the board, the results are quite similar for the literal, idiomatic and random data.
5. The paper ends with an analysis of 1) the frequency of idiomatic translations to show that the upweighting and knn-MT improve performance for both low- and high-frequency idioms, and 2) an error analysis, where "major" and "severe" errors are annotated

**Questions For The Authors:**

- Could you elaborate on the procedure used to label PIE usage in Opensubtitles as idiomatic or literal? How many annotators were involved, were they native speakers of English and Japanese/Finnish/French? What was the inter-annotator disagreement level? UPDATE AFTER REBUTTAL: apologies for missing the comment about speakers being native in the paper
- Could you elaborate on what, in your opinion, we can learn about real idioms and how frequent they need to be based on the synthetic experiment that you conducted in light of the simplifications I pointed out above?
- Could you reflect on the extent to which the upweighting and knn-MT methods are idiom-specific?

**Reasons To Accept:**

- The paper presents a very useful dataset of idiomatic translations for French, Finnish and Japanese. As this paper points out, and as previous papers have done, NMT systems still struggle with idioms, partially due to their low frequency. The majority of the resources currently used is in English, and having data for these three additional languages could support analyses in future work.
- The paper presents evidence for the role of frequency in the acquisition of idiomatic translations, both in a synthetic experiment and using real data. While the role of frequency has been assumed by previous work, few articles make this explicit, which is a meaningful contribution to have for the subcommunity interested in figurative language processing in NLP.

**Reasons To Reject:**

- Regarding the synthetic experiment: It is impossible to tell to what extent the findings from the artificial language translation experiment generalise to natural data, where non-compositional translations are much more complex. To name 3 reasons: 1) idioms have various conventionalities (~ratio between idiomatic vs literal meaning) and the frequency above which models default to the non-compositional translation likely interacts with conventionality, 2) words and n-grams contained in the idioms themselves can appear outside of the idiom, 3) idioms require disambiguation unless the idiomatic meaning is 100% conventional, 4) many idiomatic translations can be partially compositional. The artificial experiment is interesting in itself, but to assume that this is a proxy for idiom processing seems like a stretch.
- Regarding the new dataset: While the newly created dataset could potentially be a very useful resource (idiom analyses are predominantly using English corpora), the paper is not very elaborate about how the authors ensure quality control for this corpus. Who annotates the Opensubtitles sentences for idiomaticity? The paper is slightly vague about this, which suggests the authors may have annotated this. When presenting a new dataset meant to be used by future work, the proper way to construct that would be using external annotators, ideally multiple annotators per example to be able to estimate reliability of the annotations and measures of disagreement. Moreover, the corpus is constructed using idioms from a few websites, instead of from idioms taken from idiom dictionaries.
- Regarding the proposed upweighing and KNN methods: For the majority of language and score combinations (see Figure 3), the impact that the methods have on idiomatic vs random data is similar; hence the proposed MT modelling methods seem far from idiom-specific. Therefore, the results simply appear to indicate that "better NMT systems are also better at idiomatic translations".
- Across the board, the paper appears to lack focus and a single, clear storyline. The experiments seem somewhat disconnected, which makes it hard to read and understand what the main contributions are.

**Reproducibility:**

4: Could mostly reproduce the results, but there may be some variation because of sample variance or minor variations in their interpretation of the protocol or method.

**Reviewer Confidence:**

4: Quite sure. I tried to check the important points carefully. It's unlikely, though conceivable, that I missed something that should affect my ratings.

**Typos Grammar Style And Presentation Improvements:**

- Table 5: Perhaps put the tables/figures at the top of the page rather than in the middle of running text?
- line 469: "they tended to contain named entities", This is something that can be easily supported with factual information: perform NER and report the relative frequencies of named entities in the sentences for which the performance did/did not drop.
- line 465: "it's" --> "it is"

---

> ### Author Rebuttal · Authors · 2023-08-29
>
> Thank you for your comments, and we’re glad that you’ve taken the time to review our manuscript thoroughly. We are glad that you appreciate the work as a contribution to figurative language processing. We wrote a general response as well, and pasted relevant sections here. See below for responses to individual questions and comments as well:
>
> ### General response
>
> We thank all reviewers for their detailed responses and suggestions. We plan to update our manuscript based on some suggestions raised, and we also respond to some common concerns below:
>
> 2. Applicability of the synthetic experiment
>
> Several reviewers also raised concerns about the applicability of the synthetic experiment to real data. We would like to reiterate that this is meant to be a simple experiment by design and is not meant to capture the complexity of idioms in natural language. The motivation behind this experiment was to provide a simple testing ground so that factors such as corpus size, idiom frequency, sentence length, and informativity of the context could be easily manipulated. Past analysis in this area has focused on real idioms already (Shao et al 2018, Dankers et al. 2022), but this is based on a few examples, and it is not as easy to manipulate natural language data. This experiment was meant to be unambiguous and simple in order to complement the other two experiments conducted on natural language data.
>
> We agree that further extensions can be made in order to make the setting more realistic, including manipulating informativity conditions further, creating multiple non-compositional rules and longer non-compositional ngrams, and manipulating the occurrence rate of tokens in non-compositional phrases outside the phrase.
>
> 3. Annotation of literal and idiomatic sentences
>
> Annotation of literal and idiomatic sentences was conducted by the authors, who had access to the idiom present in that sentence, as well as the literal and figurative translations of that idiom in the target language (english). As stated in section 6.3, the authors were fluent in Japanese and French, but not Finnish.
>
> We will validate the literal/idiomatic judgments with external annotators in the camera-ready version, and report inter-rater reliability.
>
> ### Comments
>
> > Regarding the synthetic experiment: It is impossible to tell to what extent the findings from the artificial language translation experiment generalise to natural data, where non-compositional translations are much more complex. To name 3 reasons: 1) idioms have various conventionalities (~ratio between idiomatic vs literal meaning) and the frequency above which models default to the non-compositional translation likely interacts with conventionality, 2) words and n-grams contained in the idioms themselves can appear outside of the idiom, 3) idioms require disambiguation unless the idiomatic meaning is 100% conventional, 4) many idiomatic translations can be partially compositional. The artificial experiment is interesting in itself, but to assume that this is a proxy for idiom processing seems like a stretch.See general response, item 2. To respond to the points more directly, the synthetic experiment is not meant to capture all the nuances of idioms in natural language, as noted in the last paragraph of section 2.2, this experiment is moreso meant to capture non-compositionality in a very controlled setting, and the current definition includes things like multi-word expressions, while not currently dealing with ambiguity in language. The artificial setting is not necessarily as restrictive as we presented it, and can be extended to model other phenomena as outlined below. As there is, to our knowledge, no existing literature on evaluating non-compositionality handling in a fully controlled context, we emphasize that we are starting with a single simple case, but the approach can be extended to capture important factors like the ones that you have mentioned:
>
> 1. Conventionality - this is true for natural language idioms, and is something that we could potentially analyze in the opensubtitles results. However, to do this we would potentially have to look at much more data than opensubtitles, since many idioms don’t appear at all in their literal form in opensubtitles. We can update the manuscript with conventionality (figurative/literal counts), but a full investigation may be outside the scope of this paper.
>
> 2. This is technically true for the one non-compositional expression as well, as the unigrams “0” and “1” also appear separately outside the phrase. The proportion of times “0” or “1” can appear outside of the phrase “0 1” can also be controlled, and we can consider adding this to the manuscript. The synthetic experiment can also be extended to include longer non-compositional phrases, but a two-token phrase is the simplest case.
>
> 3. Yes, this is true in many cases. In the opensubtitles data, we can also add conventionality based on counts for each idiom as stated in point 2. Because the datasets generated are fully controllable, the informativity of the context (ease of disambiguation) can also be manipulated directly, since P(idiomatic translation|ctx) can be calculated directly.
>
> 4. This is true, but to reduce ambiguity, we treat partially compositional phrases as non-compositional since they cannot be completely reconstructed. We agree that this is an important factor to consider in future work though.
>
> > Regarding the new dataset: While the newly created dataset could potentially be a very useful resource (idiom analyses are predominantly using English corpora), the paper is not very elaborate about how the authors ensure quality control for this corpus. Who annotates the Opensubtitles sentences for idiomaticity? The paper is slightly vague about this, which suggests the authors may have annotated this. When presenting a new dataset meant to be used by future work, the proper way to construct that would be using external annotators, ideally multiple annotators per example to be able to estimate reliability of the annotations and measures of disagreement. Moreover, the corpus is constructed using idioms from a few websites, instead of from idioms taken from idiom dictionaries.
>
>
> Thank you for raising this concern. The authors did annotate whether or not a sentence contained idiomatic statements with respect to the reference translation, while having access to a literal translation of the idiom as well as the idiomatic translation in English. In order to validate the literal/idiomatic split, we plan to hire external annotators to annotate a subset, and measure agreement between our judgments and the external annotators.
>
> For data, our data sources do already include Wikitionary (for Finnish) and Kotozawa (a japanese dictionary of proverbs and idioms). However, language learning sites also contain some idioms that are not present in dictionaries. A list of data sources is available in the linked repository under `metaphor-translation/data/external/idiom_sources.txt`. If you have suggestions for online idioms dictionaries for any of our languages, please let us know.
>
> > Regarding the proposed upweighing and KNN methods: For the majority of language and score combinations (see Figure 3), the impact that the methods have on idiomatic vs random data is similar; hence the proposed MT modelling methods seem far from idiom-specific. Therefore, the results simply appear to indicate that "better NMT systems are also better at idiomatic translations".
>
> Although it is true that a better NMT model will likely be better at idiomatic translation, some idiom-specific methods such as sentence weighting improve this further, given a base model. In Table 4, upweighting on its own does improve translation of idiomatic sentences. We expected kNN-MT to benefit idiomatic sentences more than random sentences, but in practice we found that the effect was more uniform across test sets. kNN-MT based on an unweighted model outperforms vanilla kNN-MT in many cases, indicating that idiom-specific upweighting does help the model even when it has been improved with kNN-MT already.
>
> > Across the board, the paper appears to lack focus and a single, clear storyline. The experiments seem somewhat disconnected, which makes it hard to read and understand what the main contributions are.
>
> We apologize if the paper is somewhat disorganized. We can make the goals of each section more clear and explicit, such as emphasizing that the synthetic experiment is about non-compositionality in a controlled environment. If you have any additional suggestions for clarity, please let us know.
>
> ### Answers to Questions
>
> > Could you elaborate on the procedure used to label PIE usage in Opensubtitles as idiomatic or literal? How many annotators were involved, were they native speakers of English and Japanese/Finnish/French? What was the inter-annotator disagreement level?
>
> See general response, item 3. In addition, the authors are fluent speakers, but not native, in French and Japanese in section 6.3.
>
> > Could you elaborate on what, in your opinion, we can learn about real idioms and how frequent they need to be based on the synthetic experiment that you conducted in light of the simplifications I pointed out above?
>
> As stated before, the synthetic experiment was not supposed to capture the nuances of real idioms, but to provide insight into non-compositional patterns in a controlled setting.
>
> > Could you reflect on the extent to which the upweighting and knn-MT methods are idiom-specific?
>
> Upweighting is very idiom specific as only potentially idiomatic phrases were upweighted, as stated in the paragraph on upweighting in section 5.
> kNN-MT is not idiom-specific, but is expected to help models more with rare patterns/entities as reported in the Khandelwal et al 2021 paper (on kNN-LM). When combined with upweighting, it does become more idiom specific as the representations used to build the datastore would be those of the model trained with upweighting.

---

### Official Review · Reviewer_ZCwa · 2023-08-09

**Soundness:** 4

**Excitement:**

4: Strong: This paper deepens the understanding of some phenomenon or lowers the barriers to an existing research direction.

**Paper Topic And Main Contributions:**

This paper does an analysis of idiomatic MT using both synthetic language and real language, including constructing a new multilingual corpus for testing the behavior.  The paper observed that very frequent idioms are learned well, but less frequent idioms are more difficult, then suggests two techniques that should help increase the visibility of less frequent idioms (up-weighting and kNN-MT).  The results and subsequent error analysis and informative and convincing that the authors are looking in the right direction. This paper does not present a full solution to problems with idiomatic translation, but is a solid step in the process.


**Questions For The Authors:**

Question A: Regarding the use of kNN-MT, was the retrieval limited to the idiomatic phrase or used for every word in the sentence (i.e., vanilla kNN-MT from Khandelwal et al., 2021)?

Question B: Regarding Figure 1, do you have thoughts why the large transformer seems to do better with the uninformed context than the informed context (with a high number of idioms)?  Note that in several cases the medium and/or small transformer does better.

**Reasons To Accept:**

- careful analysis of problem in existing models

- theoretical study with synthetic data pointing toward solution, then use of simple techniques to provide the direction of a solution

- lots of analysis and multiple evaluations of results

**Reasons To Reject:**

none


**Reproducibility:**

4: Could mostly reproduce the results, but there may be some variation because of sample variance or minor variations in their interpretation of the protocol or method.

**Reviewer Confidence:**

4: Quite sure. I tried to check the important points carefully. It's unlikely, though conceivable, that I missed something that should affect my ratings.

**Typos Grammar Style And Presentation Improvements:**

- line 13: tilde is above 4 in 4k, did you mean ~4k?

- line 95: in improves -> it improves

- equation 3: The notation after the alpha is very hard to interpret due to super/subscripting.  After reading the text (lines 333-338), I believe the alpha term is being raised to the power of 1 if x(i) is in A and 0 if not?  So this is an indicator function for A?  Now I can see it, but I suggest changing it for easier interpretation.

---

> ### Author Rebuttal · Authors · 2023-08-29
>
> Thank you for your review, we are glad that you appreciate the methods used in our study! See below for answers to questions:
>
> ### Answers to Questions
>
> > Question A: Regarding the use of kNN-MT, was the retrieval limited to the idiomatic phrase or used for every word in the sentence (i.e., vanilla kNN-MT from Khandelwal et al., 2021)?
>
> kNN-MT was used in the usual way, for every word in the sentence. It may be possible to use kNN-MT only on idiomatic phrases for increased efficiency, but we leave this for future work.
>
> > Question B: Regarding Figure 1, do you have thoughts why the large transformer seems to do better with the uninformed context than the informed context (with a high number of idioms)? Note that in several cases the medium and/or small transformer does better.
>
> I am not sure if this is a general pattern, can you point out the cells that you see this pattern in and maybe I can comment more specifically? In general though, the large transformer tended to have more variance in training runs, especially with a larger number of idioms. This may be due to overfitting which gets worse with a greater number of non-compositional examples, but I’m also not completely sure.

---

### Official Review · Reviewer_3Wyz · 2023-08-12

**Soundness:** 3

**Excitement:**

4: Strong: This paper deepens the understanding of some phenomenon or lowers the barriers to an existing research direction.

**Missing References:**

-  The reference to Unbabel (2019) lacks a URL
- Some papers are cited without the conference or journal they were accepted in, in the reference section. The authors could eventually complete those references.

**Paper Topic And Main Contributions:**

This paper describes methods for improving the idiom translations generated by transformer-based machine translation systems. Idiomatic expressions are challenging for machine translation systems because they often cannot be translated word-by-word and they may occur rather infrequently in training data. The paper approach is inspired by Kandpal et al. (2022) analysis of the long-tail knowledge learning issues shown by language models : it studies the effect of retrieval augmentation and training objective modification on the quality of idiomatic expressions translations.

After an initial study of the ability of encoder-decoder models to learn a non compositional rule occurring with various frequencies in the training data using synthetic data, the main experiments are ran on a created resource of 4k natural sentences containing idiomatic expressions in French, Finnish and Japanese, extracted from the OpenSubtitles corpus.

First the authors evaluate two commercial systems on the created dataset, and then evaluate their approach.
Modifying a recent strong translation model (Ma et al., 2021), the authors evaluate (1) the effect of the up-weighing of the training loss of sentences containing potential idiomatic sentences, (2) the effect of augmenting the model with a retrieval component (Khandelwal et al., 2021) and (3) and the combined effects of (1) and (2) on the quality of the translation of the idioms. Evaluations are conducted with both automatic scoring metrics and human ratings.
(3) is found to improve the translation of sentences containing idioms by an average of 10.4% in absolute accuracy, with human ratings, while not harming the performance the model for other sentences. The authors show the effectiveness of the proposed methods for idioms occurring with various relative frequencies in the parallel training data.

The authors shared the code and data created for the experiments.


**Questions For The Authors:**

- A. Can you add references for the up-weighting approach?
- B. Can you add references for related synthetic data experiments?
- C. What method is used to lemmatize the idioms?
- D.  The best softmax temperature values and number of neighbours retrieved per query parameters of the Khandelwal et al., 2021 model are not all in your parameter search space, can you explain how you picked this hyperparameter search space?
- E. Would it be possible to share the results obtain on the random set coming from the OpenSubtitles dataset that seems to be in the additional material?
- F. How did you proceed to label the literal instances of the resource?

**Reasons To Accept:**

- The paper is very clearly written, the level of details provided is mostly sufficient, the additional material (code and data) is also a clear contribution, and the references given are solid and interesting. There is enough new material for a long paper.
- Retrieval augmentation has never been used in relation to idiomatic expressions processing to my knowledge, and is a promising approach.
- Although I thought this experiment was too limited, experimenting using synthetic data for studying of compositional rules learning is an inspiring idea (I may be unaware of previous work using similar method, cf Questions for Authors).
- The experiments show that the method improves the translation of idiomatic expressions.



**Reasons To Reject:**

A. Contextual translation : line 053 "However, we would ideally want to make use of the contextual translation abilities of neural models."
- The authors do not really deal with potential idiomatic expressions that can be translated word-by-word or non compositionally depending on the context. They focus instead on idioms that are, in their large majority, impossible to translate word-by-word. The word-by-word translations are separated manually and form a small literal set for each language. It would be interesting to extend the experiments to more ambiguous expressions.
- The authors could eventually distinguish word-by-word from literal translations : if the same idiom is understandable in two languages it can be both figurative and word-by-word (page 3, line 197 says that literal instances of the test set only contains literal uses of the expressions, but it is not the case in the fr final literal test set, e.g. "Moi, j'ai bien une idée... Vous donnez votre langue au chat ? Couper le courant.").
- It would be beneficial to study the diversity of translations found in the OpenSubtitles corpus for each given idiom. It would be informative to know if they are always translated the same way or if there is a high variability for this study.

B. Created resource from Open Subtitles (Lison et al., 2018)
- Some subtitles in this dataset are generated by machine translation systems. The authors do not mention this fact, so I am not sure that they filter them out. If they do I don't know if the metadata indicating which subtitles have been generated automatically are always reliable.
-  I can read the created dataset in French as a native speaker, and the idiom matching function could be improved. It currently doesn't match the word boundaries ( e.g. The entry : "rendre le âme, I take people's souls before they die., JE prends les âmes des gens avant qu'ils meurent." matches the verb "prendre" as an instance of the verb "rendre")


C. The synthetic data framework is interesting but very simplified. I would like to see what happens with more than one compositional rule and what happens when the context decides if the expression must be translated literally or metaphorically.

D. Evaluation
- The sizes of the test sets of the "literal" translations in the 3 languages are a little small to reach a strong significance in the evaluation.
- In the files shared by the authors, there is a random test set of the OpenSubtitles corpus. I would be curious to know the automatic metrics scores on this one, it would help in the interpretation of the overall results of the experiments. Domain adaptation is interesting but I think that having results on a random set drawn from the same distribution is important here.



**Reproducibility:**

5: Could easily reproduce the results.

**Reviewer Confidence:**

3: Pretty sure, but there's a chance I missed something. Although I have a good feel for this area in general, I did not carefully check the paper's details, e.g., the math, experimental design, or novelty.

**Typos Grammar Style And Presentation Improvements:**

- Some information of section 6.2 about the OpenSubtitles corpus size would be interesting in Section 3.
- Page 3, line 198, I struggled to understand the sentence: " The first 100 examples containing each idiom’s lemmatized form were collected, and up to the first 3 (for Japanese) or 5 (for Finnish and French) literal and figurative examples in this set were collected.".  Is "to create the test set" missing at the sentence end?
- Page 11, line 785  : "without context" should be "without informative context"?
- Page 11 line 789 : "15, 15 and 25" should be "15, 20 and 25"?

---

> ### Author Rebuttal · Authors · 2023-08-29
>
> Hello, thank you for reading the paper so carefully and providing many suggestions! We’re glad that you appreciated the new resources as well as the synthetic experiment! We wrote a general response as well, and pasted relevant sections here. See below for responses to individual questions and comments as well:
>
> ### General response
>
> We thank all reviewers for their detailed responses and suggestions. We plan to update our manuscript based on some suggestions raised, and we also respond to some common concerns below:
>
> 1. In-domain random sentences set
>
> Several reviewers have raised the fact that seeing the results for an in-domain set of randomly selected sentences would also be helpful. We agree with this. We initially thought that we could fulfill two criteria at once by choosing an out-of-domain random set, but we agree it is also important to see the in-domain results.
>
> Below are automatic metrics results for a random test set (same size as the one in the paper), selected from the dev set of our opensubtitles data. We used the same process as in the paper to create the random test set, except using opensubtitles sentences. Unfortunately we don’t have access to the Japanese base model checkpoint right now, but we will also update this result with Japanese afterward, as well as conduct human evaluation on all three languages.
>
> |  language | metric | base | upweight | knn | upweight+knn |
> |---|---|---|---|---|---|
> | FI | BLEU | 0.2056 | 0.2044 | 0.1991 | **0.2120** |
> |  | RougeLSum | 0.5044 | 0.5048 | 0.5067 | **0.5224** |
> |  | Meteor | 0.4739 | 0.4728 | 0.4732 | **0.4880** |
> |  | Bertscore | 0.9321 | 0.9324 | 0.9316 | **0.9351** |
> | FR | BLEU | 0.3197 | 0.3079 | 0.3142 | **0.3258** |
> |  | RougeLSum | 0.5942 | 0.5884 | 0.5949 | **0.6036** |
> |  | Meteor | 0.5803 | 0.5715 | 0.5774 | **0.5865** |
> |  | Bertscore | 0.9443 | 0.9427 | 0.9438 | **0.9461** |
>
> 2. Applicability of the synthetic experiment
>
> Several reviewers also raised concerns about the applicability of the synthetic experiment to real data. We would like to reiterate that this is meant to be a simple experiment by design and is not meant to capture the complexity of idioms in natural language. The motivation behind this experiment was to provide a simple testing ground so that factors such as corpus size, idiom frequency, sentence length, and informativity of the context could be easily manipulated. Past analysis in this area has focused on real idioms already (Shao et al 2018, Dankers et al. 2022), but this is based on a few examples, and it is not as easy to manipulate natural language data. This experiment was meant to be unambiguous and simple in order to complement the other two experiments conducted on natural language data.
>
> We agree that further extensions can be made in order to make the setting more realistic, including manipulating informativity conditions further, creating multiple non-compositional rules and longer non-compositional ngrams, and manipulating the occurrence rate of tokens in non-compositional phrases outside the phrase.
>
> 3. Annotation of literal and idiomatic sentences
>
> Annotation of literal and idiomatic sentences was conducted by the authors, who had access to the idiom present in that sentence, as well as the literal and figurative translations of that idiom in the target language (english). As stated in section 6.3, the authors were fluent in Japanese and French, but not Finnish.
>
> We will validate the literal/idiomatic judgments with external annotators in the camera-ready version, and report inter-rater reliability.
>
> ### Comments
> #### Comment A
> > The authors do not really deal with potential idiomatic expressions that can be translated word-by-word or non compositionally depending on the context. They focus instead on idioms that are, in their large majority, impossible to translate word-by-word. The word-by-word translations are separated manually and form a small literal set for each language. It would be interesting to extend the experiments to more ambiguous expressions.
>
> Thank you for pointing this out. We agree that there are many expressions which can be both idiomatic and ambiguous. This was based on idioms recorded in language-learning sites, which tend towards idioms that are not possible to translate word-for-word, as these may be expressions that language learners are more likely to search for. More ambiguous expressions were not focused on in this work, but we agree that this is an interesting avenue for future work.
>
> > The authors could eventually distinguish word-by-word from literal translations : if the same idiom is understandable in two languages it can be both figurative and word-by-word (page 3, line 197 says that literal instances of the test set only contains literal uses of the expressions, but it is not the case in the fr final literal test set, e.g. "Moi, j'ai bien une idée... Vous donnez votre langue au chat ? Couper le courant.").
>
> Thanks for catching this example. We agree that some idioms are understandable in both source and target languages, perhaps because the same idiom exists in both languages, or because the expression depends on common-sense knowledge. For this particular example, the translation given as a reference was “Okay, okay, so, then, let me throw this out there and see if the cat licks it up”. Since the translation would be matched against this translation, it was categorized as literal since it didn’t contain a paraphrase of “I give up”. This is not the best translation of this sentence in general, but we chose to just categorize based on english translations for consistency.
>
> > It would be beneficial to study the diversity of translations found in the OpenSubtitles corpus for each given idiom. It would be informative to know if they are always translated the same way or if there is a high variability for this study.
>
> Thank you for this suggestion, this is something that we can add to the camera-ready version. We were also concerned about the potential for multiple valid translations, so we emphasize that the human evaluation also considered semantically similar phrases as correct, e.g. for a reference of “This is just way over my head”, “this is confusing” is marked as correct.
>
>
> #### Comment B
> > Some subtitles in this dataset are generated by machine translation systems. The authors do not mention this fact, so I am not sure that they filter them out. If they do I don't know if the metadata indicating which subtitles have been generated automatically are always reliable.
>
> Thanks for pointing this out as well! We used the version of opensubtitles from the huggingface datasets library, which does not have this metadata. However, we can take a look at the metadata and see if it is reliable/covers our languages sufficiently. If so, we can also divide the results by references that are machine translated or not.
>
> > I can read the created dataset in French as a native speaker, and the idiom matching function could be improved. It currently doesn't match the word boundaries ( e.g. The entry : "rendre le âme, I take people's souls before they die., JE prends les âmes des gens avant qu'ils meurent." matches the verb "prendre" as an instance of the verb "rendre")
>
> Oh, thank you for pointing this out. We can definitely improve the matching and exclude anything in the current test sets that doesn't match on word boundaries. The current sorting into idiomatic/literal was done by an author and an attempt was made to exclude these cases, but we may have missed a few sentences like this.
>
> #### Comment C
> > The synthetic data framework is interesting but very simplified. I would like to see what happens with more than one compositional rule and what happens when the context decides if the expression must be translated literally or metaphorically.
>
> See general response, item 2. I think exploring ambiguous expressions and multiple non-compositional rules would be interesting for future work.
>
> #### Comment D
> > The sizes of the test sets of the "literal" translations in the 3 languages are a little small to reach a strong significance in the evaluation.
>
> We agree that the literal test sets are relatively small, but nonetheless we conducted significance tests, which are covered in appendix E and F. Significance (\alpha \leq 0.05) was reached for the fi literal set, but not for the other two. It is a bit hard to expand the literal test sets because most of the idioms that we collected do not have a plausible literal interpretation, or the literal interpretation does not appear in OpenSubtitles.
>
> #### Comment E
>
> > In the files shared by the authors, there is a random test set of the OpenSubtitles corpus. I would be curious to know the automatic metrics scores on this one, it would help in the interpretation of the overall results of the experiments. Domain adaptation is interesting but I think that having results on a random set drawn from the same distribution is important here.
>
> See general response, item 1.
>
> ### Answers to Questions
>
> > A, B: Can you add references for the up-weighting approach? Can you add references for related synthetic data experiments?
>
> Thank you for the suggestion, we plan to add these references for instance weighting in MT:
>
> - Instance Weighting for Neural machine translation domain adaptation, Wang et al. 2017
> - Discriminative Instance Weighting for Domain Adaptation in Statistical Machine Translation, Foster at al. 2010
>
> For the synthetic data experiment, we are not aware of any similar experiments in MT. There are experiments on memorization of idioms in the Dankers et al 2022 and Haviv et al 2022 papers cited in the manuscript, but these are on english idioms.
>
> > C: What method is used to lemmatize the idioms?
>
> The stanza english lemmatizer was used. We will add a citation to stanza in the updated manuscript.
>
> > D: The best softmax temperature values and number of neighbours retrieved per query parameters of the Khandelwal et al., 2021 model are not all in your parameter search space, can you explain how you picked this hyperparameter search space?
>
> The use case in Khandelwal et al 2021 is quite different from ours, although we were inspired by that paper. Since many idioms occur relatively rarely in the training set, we set the max number of neighbours to be 20, rather than 64 (the default in the original kNN-MT paper). Other hyperparameters were just set to reasonable grid searches, such as $\lambda$ and temperature, as outlined in Appendix C.
>
> > E: Would it be possible to share the results obtain on the random set coming from the OpenSubtitles dataset that seems to be in the additional material?
>
> See general response, item 1.
>
> > F. How did you proceed to label the literal instances of the resource?
>
> See general response, item 3.

---

### Meta-Review · Area_Chair_BVDS · 2023-09-18

**Recommendation:** 4

**Metareview:**

Four reviewers gave the Soundness scores 3/4/2/4 and Excitement Scores 4/4/2/3. The authors have responded to all four reviewers and all reviewers acknowledged the author responses.
All reviewers recognise the contribution of this paper on addressing idiomatic MT, one of the bottlenecks in MT field, especially  in French, Finnish, and Japanese.
Majority of the reviewers suggest this is a good and solid work on Soundness, but one reviewer raised many concerns regarding the synthetic experiment, dataset quality control, methods justification, and paper coherence.
There are active discussions between reviewers and authors, especially on comments from the lower review scores.
Overall this is an interesting paper with solid work. It shall appear in main / findings.
Appendix content can be reorganised to save spaces and reduce number of pages. I hope the authors can address this.

---

### Decision · Program_Chairs · 2023-10-07

**Decision:**

Accept-Main

**Comment:**

Four reviewers gave the Soundness scores 3/4/2/4 and Excitement Scores 4/4/2/3. The authors have responded to all four reviewers and all reviewers acknowledged the author responses.
All reviewers recognise the contribution of this paper on addressing idiomatic MT, one of the bottlenecks in MT field, especially  in French, Finnish, and Japanese.
Majority of the reviewers suggest this is a good and solid work on Soundness, but one reviewer raised many concerns regarding the synthetic experiment, dataset quality control, methods justification, and paper coherence.
There are active discussions between reviewers and authors, especially on comments from the lower review scores.
Overall this is an interesting paper with solid work. It shall appear in main / findings.
Appendix content can be reorganised to save spaces and reduce number of pages. I hope the authors can address this.